# Recent Technological and Intellectual Property Trends in Antibody–Drug Conjugate Research

**DOI:** 10.3390/pharmaceutics16020221

**Published:** 2024-02-03

**Authors:** Youngbo Choi, Youbeen Choi, Surin Hong

**Affiliations:** 1Department of Safety Engineering, Chungbuk National University, Cheongju 28644, Chungbuk, Republic of Korea; ybc@cbnu.ac.kr; 2Department of BigData, Chungbuk National University, Cheongju 28644, Chungbuk, Republic of Korea; 3Department of Biotechnology, CHA University, Pocheon 11160, Gyeonggi, Republic of Korea; boostburst@naver.com

**Keywords:** antibody–drug conjugates (ADCs), technology trends, intellectual property (IP) landscape, market, key players, innovation

## Abstract

Antibody–drug conjugate (ADC) therapy, an advanced therapeutic technology comprising antibodies, chemical linkers, and cytotoxic payloads, addresses the limitations of traditional chemotherapy. This study explores key elements of ADC therapy, focusing on antibody development, linker design, and cytotoxic payload delivery. The global rise in cancer incidence has driven increased investment in anticancer agents, resulting in significant growth in the ADC therapy market. Over the past two decades, notable progress has been made, with approvals for 14 ADC treatments targeting various cancers by 2022. Diverse ADC therapies for hematologic malignancies and solid tumors have emerged, with numerous candidates currently undergoing clinical trials. Recent years have seen a noteworthy increase in ADC therapy clinical trials, marked by the initiation of numerous new therapies in 2022. Research and development, coupled with patent applications, have intensified, notably from major companies like Pfizer Inc. (New York, NY, USA), AbbVie Pharmaceuticals Inc. (USA), Regeneron Pharmaceuticals Inc. (Tarrytown, NY, USA), and Seagen Inc. (Bothell, WA, USA). While ADC therapy holds great promise in anticancer treatment, challenges persist, including premature payload release and immune-related side effects. Ongoing research and innovation are crucial for advancing ADC therapy. Future developments may include novel conjugation methods, stable linker designs, efficient payload delivery technologies, and integration with nanotechnology, driving the evolution of ADC therapy in anticancer treatment.

## 1. Background of ADC Therapeutic Technology

The antibody–drug conjugate (ADC) therapy technology is a next-generation therapeutic approach to overcome the limitations of conventional cancer chemotherapy. It is considered one of the next-generation anticancer treatment technologies that leverage the targeted selectivity of antibodies and the cell-killing efficacy of cytotoxic drugs to enhance therapeutic effects while minimizing side effects [1,2,3,4]. This technology involves the use of a drug composed of a low-molecular-weight cytotoxic agent (chemotherapeutic drug) chemically linked to an antibody that interacts with a specific antigen overexpressed on the surface of cancer cells through a chemical linker (Figure 1). This structure allows for the targeted delivery of the cytotoxic drug to cancer cells, enhancing the effectiveness of the anticancer treatment while minimizing adverse effects.

The optimal ADC therapy is characterized by its ability to maintain stability in the bloodstream, accurately reach targeted cancer cells, and ultimately release the cytotoxic payload in close proximity to the specified cancer cells for effective treatment. Essential components of ADCs in achieving these objectives encompass tumor-targeting antibodies designed to correspond to antigens expressed on cancer cells, along with linkers and cytotoxic payloads. The conjugation methods employed for these components represent a critical technological aspect in ADC manufacturing, enabling the precise assembly of these elements and ensuring optimal therapeutic outcomes.

### 1.1. Selection of Target Antigens

The target antigen expressed on cancer cells serves as the navigation system for ADC therapy, determining the mechanism for recognizing cancer cells and delivering the cytotoxic payload. The selection of an ideal target antigen is the first crucial consideration in this process. The criteria for the ideal selection of a target antigen typically involve its overexpression in cancer cells while being rare or very lowly expressed in normal tissues. Additionally, the antigen should be expressed on the surface of cancer cells. It is also essential that the chosen antigen is not secreted in the bloodstream to avoid unwanted binding of ADCs in undesired locations. Currently developed ADC therapies have selected target antigens such as HER2, trop2, nectin4, and EGFR for solid tumors and CD19, CD22, CD33, CD30, BCMA, and CD79b for hematologic malignancies [1,5,6]. These antigens have been chosen based on their overexpression in cancer cells and their suitability for effective ADC therapy. 

### 1.2. Cancer Cell-Targeting Antibodies

Antibodies targeting cancer cells play a pivotal role in facilitating specific binding between the target antigen and ADCs. These antibodies should demonstrate high binding affinity to the target antigen, low immunogenicity, and an extended half-life. In the initial stages of ADC therapy development, antibodies derived from mice were commonly utilized. However, due to severe immunogenic side effects, especially associated with murine antibodies, the prevailing trend now predominantly favors the use of humanized antibodies produced through recombinant technology [7,8,9]. Humanized antibodies are generated by incorporating key regions of the mouse-derived antibody into a human antibody framework. This approach preserves the specificity and high binding affinity of the mouse antibody while minimizing the risk of immune reactions in humans. The transition towards humanized antibodies has significantly contributed to enhancing the safety and efficacy of ADC therapies.

### 1.3. Linkers

The linker in ADCs plays a crucial role in bridging the antibody and the cytotoxic drug, representing a critical determinant of ADC stability and the profile of payload drug release. This, in turn, significantly influences therapeutic efficacy. An ideal linker should avoid inducing ADC aggregation, prevent premature payload release in the bloodstream, and facilitate the release of active drugs precisely at the desired target. Linkers are broadly classified into two main types based on cellular metabolism processes [10,11,12,13]: cleavable linkers and non-cleavable linkers. Cleavable linkers are further subdivided into chemical cleavage linkers and enzyme cleavage linkers. These linkers offer the advantage of precisely releasing cytotoxic drugs, taking into account systemic circulation and environmental disparities between normal cells and cancer cells. On the contrary, non-cleavable linkers are connected as amino acid residues within the breakdown products of the antibody, displaying low activity in the general chemical and enzymatic environments within the body, ensuring high plasma stability. Typically, non-cleavable linkers rely on enzyme hydrolysis of the ADC’s antibody component, primarily facilitated by proteases, culminating in the release of the payload in a complex form.

### 1.4. Cytotoxic Payloads

The cytotoxic payload is the component of ADCs that signifies the drug’s cytotoxic effect upon penetration into cancer cells. Given that only approximately 2% of ADCs can reach the targeted tumor site after intravenous administration, it is imperative to employ a highly effective compound as the payload. This compound should demonstrate stability under physiological conditions and possess functional groups capable of binding to antibodies. Currently, cytotoxic payloads used in ADCs primarily include potent tubulin inhibitors, DNA-damaging agents, and immunomodulators [14,15]. These compounds are selected for their capacity to exert a robust therapeutic effect within cancer cells. Tubulin inhibitors disrupt microtubule dynamics, affecting cell division; DNA-damaging agents induce DNA damage to inhibit cell proliferation, and immunomodulators modulate the immune response within cancer cells. The meticulous choice of the cytotoxic payload is crucial for attaining the desired therapeutic outcomes in ADCs [16,17,18,19] (Table 1).

### 1.5. Conjugation Methods

In addition to the selection of antibodies, linkers, and payloads, the method by which the small-molecule component (e.g., linker plus payloads) is attached to the antibody is a crucial element in the successful construction of ADCs [20,21,22]. Antibodies typically contain residues for binding reactions, such as lysine and cysteine residues. In the early development of ADC drugs, conventional coupling methods often used existing lysine or cysteine residues on the antibody through appropriate coupling reactions [23,24]. One of the most commonly used methods for connecting the payload to the lysine residues of the antibody is through the amide coupling reaction, using an active carboxylic acid ester [25,26,27]. However, the abundant presence of lysine complicates the control of site selectivity, resulting in challenges such as premature payload release and the potential for off-target toxicity.

To address these limitations, innovative strategies for ADCs, including site-specific conjugation methods, are currently in development [28,29,30,31]. Site-specific conjugation methods present a groundbreaking approach in ADC development, aiming to precisely attach the payload at specific locations and overcome challenges associated with traditional coupling methods. For example, the introduction of engineered reactive cysteine residues selectively inserted at specific positions enables precise conjugation at that site, enhancing the homogeneity of ADCs and providing tunable reactivity through the alteration of the modification site [32,33,34]. In enzymatic conjugation methods, a variety of enzymes, such as bacterial-derived formyl glycine-generating enzymes, transglutaminases, glycotransferases, and sortases, have been utilized for tag-free antibody modification techniques [35,36,37,38,39]. The reaction sites of antibodies are designed to specifically interact with the corresponding functional groups, facilitating site-specific conjugation in enzymatic methods. The incorporation of unnatural amino acids with bioorthogonal groups is also employed in site-specific conjugation [40,41,42,43]. The most common method of incorporation involves engineering transfer RNA synthetases to recognize the unnatural amino acids, resulting in the genetic coding of these unconventional building blocks. Enzymatic conjugation methods provide precise control over the site of conjugation, reducing heterogeneity and enhancing the therapeutic index of ADCs. Tag-free techniques, particularly those based on enzymatic modification, often yield conjugates with reduced immunogenicity.

## 2. Trends in Biopharmaceutical Market and Development of ADC Therapies

### 2.1. Global Trends in Biopharmaceutical Market

The global pharmaceutical market continues to grow, with a projected compound annual growth rate (CAGR) of approximately 6.9%, increasing from around USD 844 billion in 2019 to an estimated USD 1.181 trillion in 2024 [44]. The advancement in biotechnological technologies has led to a trend of focusing more on the development of biopharmaceuticals rather than synthetic drugs. Among these, the market for monoclonal antibody drugs is particularly noteworthy. The share of biopharmaceuticals in the overall pharmaceutical market has steadily increased, rising from 18% (USD 129 billion) in 2010 to 28% (USD 243 billion) in 2018. It is anticipated to continue growing, reaching an estimated 32% with a CAGR of 8.5% by 2024 [45]. The global biopharmaceutical market is led by the United States, accounting for 61% of sales in 2020. The major European countries (Germany, France, Italy, the UK, Spain) collectively hold a 17% share, while Asian countries such as Japan (5%) and China (3%) rank within the top five in market share. Within the pharmaceutical market, the oncology sector is recognized as the largest, driven by an increasing incidence of cancer worldwide [46].

According to the International Agency for Research on Cancer (IARC), global cancer cases reached 19.3 million in 2020, projected to rise by 47% to 28.4 million by 2040 [47]. In response to this growing cancer prevalence, global spending on oncology drugs, as estimated by IQVIA, is expected to grow at a CAGR of 9~12% from USD 164 billion in 2020 to USD 269 billion in 2025 [48]. Specifically, the ADC market witnessed substantial growth, reaching USD 7.35 billion in 2022, reflecting a 34.9% increase compared to USD 5.45 billion in the previous year [49]. The ADC market is projected to continue its robust growth at an average annual rate of 25.4%, reaching a total revenue of USD 28.53 billion by 2028.

### 2.2. Global Trends in Approval and Development of ADC Therapies

Innovative concepts and the careful design of chemical linkers for the creation of payload-conjugated therapies have led to significant advancements. In the year 2000, the U.S. Food and Drug Administration (FDA) granted approval for the first ADC therapy, Mylotarg^®^, intended for patients with acute myeloid leukemia [19]. This milestone marked the initiation of the ADC therapy market dedicated to cancer treatment. As of December 2021, a total of 14 ADC therapies have received global approval for the treatment of solid tumors and hematologic cancers. Presently, there are over 100 ADC candidates in various stages of clinical trials [19]. The evolution of the conceptualization and development processes of ADC therapies over the past century, dating back to 1910, is visually depicted in the diagram [1], as illustrated in Figure 2.

The approval status of ADC therapies for the treatment of hematologic malignancies is outlined below. Commencing with Adcetris, designed to target CD30 and granted approval in 2011, a cumulative total of seven therapies for hematologic cancers have received approval. Notable manufacturers include Seagen, GSK, and Pfizer, among others. The payload substances employed encompass Monomethyl auristatin E/F, Calicheacmicin, and others, as detailed in Table 2.

Furthermore, following the approval of Kadcyla, which targets HER2 for solid tumors, in 2013, seven ADC therapies have received approval. Prominent manufacturers involved in these approvals include Roche, Daiichi Sankyo, Immunomedics, Seagen, and others [49]. The intellectual property (IP) landscape of these major manufacturers will be delved into in more detail in the subsequent sections, as presented in Table 3.

Specifically, Kadcyla has exhibited effective therapeutic outcomes in HER2-positive breast cancer by selectively delivering the drug to tumor cells, thereby minimizing its impact on surrounding normal cells [50,51]. Adcetris, employed in Hodgkin’s lymphoma and CD30-positive non-Hodgkin’s lymphoma, demonstrates notable efficacy against specific positive tumors while concurrently reducing the impact on normal cells [50,52]. However, severe side effects may manifest in some patients, and treatment effectiveness could vary based on the cancer type. Enhertu has proven effective in treating various cancer types, including HER2-positive breast cancer, indicating a considerable potential for diverse cancer treatments. Nevertheless, some patients may experience moderate to severe side effects, and the specificity for HER2-negative tumors may be compromised [50,53,54].

Significantly, since its approval by the U.S. FDA in 2013, Kadcyla has been administered to HER2-positive breast cancer patients, delivering effective therapeutic outcomes by amalgamating the advantages of conventional antibody treatments and chemotherapy [50,51]. The targeted delivery of the drug resulted in a significant reduction in tumor size and improved overall survival. However, common drawbacks of ADC therapies, such as potential side effects and the development of drug resistance in some patients, persist. Ongoing research and development endeavors aim to enhance efficacy and minimize adverse effects in the realm of ADC therapy [16,22,55].

The majority of ADC developers are embracing collaborative development strategies to exchange technology and resources, facilitating mutual expansion, particularly in the field of oncology therapies. The noteworthy partnerships and collaborations poised for significance in 2022 and 2023 are outlined in Table 4.

Table 5 exhibits the summary of key patents for 11 FDA-approved ADC drugs. Among the ADC drugs approved by the FDA, four drugs (Adcetris, Polivy, Padcev, and Tivdak) utilized vedotin, comprising the MC-VC-PABC linker and MMAE cytotoxin, as outlined in US Patent 7,659,241 filed in 2003. Blenrep employed the MMAF cytotoxin (US Patent 7,662,387 filed in 2004), an analog of MMAE found in vedotin. On the other hand, ozogamicin was used as a linker cytotoxin in Mylotag and Besponsa drugs. In 2013, two patents, US Patent 10,195,288 and 9,028,833, were filed for the camtothecin-based deruxtecan and govitecan used in Enhertu and Trodelvy drugs [61].

Notably, while the four ADC drugs employed the same vedotin, they are each covered by distinct patents for their unique ADC formulas, which include different antibodies and target antigens. The filing years for these formula patents span a decade, with the earliest being US Patent 7,659,241 filed in 2004 and the latest being US Patent 10,617,764 filed in 2014. Additionally, the Mylotag and Besponsa drugs, both based on ozogamincin, were each protected by different formula patents (US Patent 5,712,374 filed in 1995 and 8,153,768 filed in 2003). Furthermore, additional patents for the ozogamin-based ADC formula were filed in 2012 and 2014.

The patent filing trends for the vedotin- and ozogamicin-based ADC drugs demonstrate that, although the individual components such as linkers, cytotoxins, and antibodies may originate from established technologies, the innovation can be achieved by developing novel ADC formulas through the combination of these elements. This strategy has the potential to advance the ADC technology and broaden its practical uses.

The general duration of patent protection is 20 years from the filing date. Therefore, it can be assumed that the patents for the ADC drugs with FDA approvals listed in Table 5 and filed before 2003 have expired as of January 2024. However, various strategies such as patent term extension, dosage patents [62], and second medical use patents [63] can be employed to secure exclusive rights beyond the initial patent duration for the ADC drugs.

For example, US Patent 7,659,241, filed on 31 July 2003, for the Adcetris drug, had its duration extended until July 15, 2026, through the patent term extension [64]. On the other hand, Roche filed a new dosage patent for the Kadcyla drug with the EPO in 2010 and received European Patent 2,459,167B1 in 2013 [62,65]. This allowed Roche to extend its exclusive rights for Kadcyla in Europe until 2030. The University of California has discovered a second medical use for an antibody produced by the hybridoma deposited as ATCC Deposit No. PTA-5817, resulting in the grant of European Patent 1,734,996B1 [63,66].

### 2.3. The Recent Clinical Status and Innovative Studies of ADCs

Between 2019 and 2022, the FDA granted approval to eight ADC drugs. In the year 2022 alone, 57 novel ADCs commenced phase 1 clinical trials, reflecting a notable 90% surge in comparison to the previous year, 2021. Additionally, the initiation of 249 new clinical trials to assess ADCs in 2022 represented a substantial 35% increase compared to the activities in 2021 [67].

Over the last two decades, substantial investments have been allocated to various components of antibody–drug conjugates (ADCs), encompassing antibodies, conjugation methods, linkers, and payloads. Conjugation methods aim to produce stable and uniform ADC products, with the field of linkers experiencing significant innovation, currently employing 33 in ongoing clinical trials. Moreover, there is notable interest in investing in over 60 novel payloads at the clinical stages of ADC development. As of 2022, merely 25% of emerging ADCs employ tubulin inhibitors as their primary mechanism of action. This trend signifies a diversification of technologies and suggests potential advancements in more effective mechanisms of action, as depicted in Figure 3.

On the contrary, a pivotal factor associated with the adverse effects of ADC therapy pertains to the premature liberation of the payload within the organism. Inadequate antigen expression on tumors leads to insufficient toxin delivery and drug resistance. Meanwhile, excessive antigen expression on normal healthy tissues causes on-target but off-tumor toxicity. Furthermore, ADC antibodies may induce immune-related adverse effects. Prevalent toxicities noted in clinical observations encompass thrombocytopenia, anemia, neutropenia, and leukopenia, with hepatotoxicity being the most predominant [68].

To address these side effects and enhance therapeutic efficacy, various emerging ADC formats have been developed, including bispecific ADCs, conditionally active ADCs known as probody–drug conjugates, immune-stimulating ADCs, and protein-degrader ADCs [69]. Each ADC format offers unique capabilities for addressing different challenges. For instance, probody therapeutics represent a novel class of recombinant antibody-based therapeutics targeting antibody activity to the tumor by exploiting the dysregulation of proteases in disease tissues. Probody–drug conjugates consist of several components, including a parental antibody, the prodomain (comprising a masking peptide linked to the N-terminus of the light chain of the parental antibody via a protease substrate), and finally, the linker/toxin [70,71]. Bispecific ADCs involve a bispecific antibody, allowing simultaneous engagement of two different targets by a single antibody-like molecule, and an ADC, facilitating the cancer-selective delivery of potent cytotoxic payloads [72,73]. Probody–drug conjugates are anticipated to improve tumor specificity, while bispecific ADCs have the potential to combat drug resistance and tumor heterogeneity. In the near future, a combination of technologies may be required to achieve the broadest therapeutic window. 

Furthermore, the development of new payloads has paved the way for innovative non-cytotoxic payloads in ADCs. Examples of such payloads include RNA inhibitors, immune agonists, and apoptosis-promoting Bcl-xL inhibitors. In particular, immune agonists, such as Toll-like receptor (TLR) agonists, stimulators of the interferon gene (STING) agonists, and glucocorticoid receptor modulators, represent a promising class of payloads aiming to achieve therapeutic effects through the precision of antibody-guided targeting with the immunomodulatory capabilities of small molecules [74,75,76]. Immune-agonist ADCs leverage antibodies to deliver immune agonists to the tumor microenvironments and release them locally. This approach is anticipated to enhance the antitumor immune response. Notably, TLRs, STING, and other immune agonists play crucial roles in modulating the immune system [77,78,79]. The preliminary clinical results of these immune-agonist ADCs have shown promising potential for a novel therapeutic approach but simultaneously underscored the necessity for meticulous investigation to confirm their safety. For example, 209P interim results for the phase I/Ib study of SBT6050, which comprised a TLR8 agonist linker payload-conjugated to a HER2-directed antibody, indicated a manageable safety profile and pharmacodynamics suggestive of myeloid, NK, and T-cell activation when SBT6050 was administered alone or in combination with pembrolizumab [80]. The phase 1 clinical trial of XMT-2056, which consisted of a HER2-targeted antibody conjugated with a STING agonist payload, was voluntarily suspended in March 2023 due to a significant Grade 5 adverse event [81]. Subsequently, the trial suspension was lifted in October 2023 after reducing the starting dose in the phase 1 dose escalation design of XMT-2056 [82]. Therefore, continued research and clinical development are imperative to comprehensively comprehend and harness the therapeutic advantages of these emerging payloads in ADCs.

## 3. Biopharmaceutical Patent Trends and ADC Therapy Technology

### 3.1. Worldwide Biopharmaceutical Patent Trends

Biopharmaceuticals are broadly classified based on the biological materials employed as active ingredients, as depicted in Figure 4. This classification includes protein therapies, cell therapies, and gene therapies. Within the realm of protein therapies, further distinctions can be made, encompassing vaccines, antibodies, and recombinant protein therapies. From a material standpoint in the patent technology classification, ADC therapies are considered part of the protein therapy technology.

To analyze patent trends in biopharmaceuticals, key keywords were identified for protein recombinants, antibodies, vaccines, and cell therapies. Subsequently, patent information was categorized based on these key terms, as detailed in Table 6. 

An in-depth analysis was conducted on the patent application trends over the past two decades at the primary patent offices of the United States (USPTO), Europe (EPO), China (CNIPA), Japan (JPO), and South Korea (KIPO). The objective was to gain insights into overall market trends and conduct a comprehensive analysis of the technological landscape in the field. Among a total of 6831 patent applications, the United States (US) constituted 38.7% of the overall applications, with Europe (EP), China (CN), and Japan (JP) holding 17.6%, 17.4%, and 17.1%, respectively. South Korea (KR) contributed 9.2% to the total application count. Figure 5 visually depicts the distribution and proportion of patent applications for biopharmaceutical technologies across the major countries’ patent offices during the last two decades.

In the examination of technology-specific distribution, the CPC classification code A61K48/00, involving medicinal preparations containing genetic material for the treatment of genetic diseases through insertion into living body cells, holds the highest number of patents, totaling 3941. Additionally, A61P35/00 (antitumor agents) and C12N15/86 (DNA related to mutations or genetic engineering, encompassing recombinant DNA technology and vectors for introducing foreign genetic material, specifically designed for host cells, especially eukaryotic cells, and virus vectors) show distributions of 2713 and 2033 patents, respectively. Following closely, A61K38/00 (medicinal preparations containing peptides) and A61K2039/505 (composition of antibodies, pharmaceutical compositions containing antigens or antibodies) exhibit distributions of 1528 and 1429 patents, respectively. This suggests the dominance of gene therapy and protein therapy technologies in the patent landscape. 

An analysis of the recent research and development activities in biopharmaceuticals reveals Switzerland leading with the highest activity at 49.5%, followed by Germany at 41.8% and South Korea at 40.3%. Recent activity serves as an indicator of changes in patent application activity over the past four years within the overall twenty-year analysis period, offering insights into the concentration of research and development based on the recent activities of patent applicants.

### 3.2. Global Patent Landscape in ADC Therapies

An analysis was conducted on patent applications and registrations pertaining to ADC therapies spanning from 1 January 2000 to 13 November 2023. The examination utilized publicly available data from the patent offices of the United States, Japan, Europe, China, and Korea. The technological classification system for ADC therapies comprises the primary category ‘antibody drug conjugates’ with sub-technologies focusing on ‘linker and payload’. A comprehensive patent trend analysis was performed, leveraging pertinent and effective patent data for each classification. 

The key terms employed for technology classification and search criteria are delineated in Table 7. In our annual analysis of patent application trends across major patent offices, we aimed to understand overall technological market trends and conduct a comparative analysis of the technological positions held by each country in the specified field. Based on a total of 6649 patent applications, the findings indicate that the United States dominates with the highest share at 28.9%, followed by China at 23.3%, Europe at 18.5%, Japan at 16.5%, and Korea at 12.9%. This analysis contributes insights into the global technological landscape and the relative positions of countries within the designated technology domain (see Figure 6).

Analyzing the annual patent application trends, a total of 6649 applications were filed over the past two decades. There is an average annual increase of 15.3%, with applications growing from 38 in 2002 to 719 in 2021. Notably, for patents simultaneously filed in three or more of the countries (United States, Europe, China, Japan, and Korea), the trend reflects the overall increase in patent applications, exhibiting an average annual growth of 15.5%, rising from 23 in 2002 to 354 in 2021. The trajectory of patent applications by major countries for ADC therapies over the years is visually presented in Figure 7.

In our analysis of the detailed distribution of ADC therapy technologies using the CPC classification, the majority of patents, totaling 4264, fall under A61P35/00 (antitumor agents). Furthermore, patents are distributed across specific categories: 3443 under A61K47/6803 (pharmaceutical preparations with characteristics in the active ingredients used: drug–antibody defined by pharmacological or therapeutic activity), 2333 under A61K47/6849 (pharmaceutical preparations with characteristics in the active ingredients used: antibodies targeting receptors, cell surface antigens, or cell surface determinants), 1890 under A61K2039/505 (composition of antibodies: pharmaceutical compositions containing antigens or antibodies, substances for immune analysis), and 1841 under A61K47/6851 (pharmaceutical preparations with characteristics in the active ingredients used: antibodies targeting determinants of tumor cells).

### 3.3. Analysis of Key Patent Applicants and Market Viability in ADC Therapy

To identify key players in the ADC therapy field, a global patent applicant analysis was conducted. As shown in Figure 8, Daiichi Sankyo Inc. leads with 277 patent filings, followed by Seattle Genetics Inc. (Bothell, WA, USA) with 253 and Genentech Inc. with 228. To analyze market viability, the ratio of triadic patents and its relevance as an indicator of major market presence was examined. The results indicate that Japan has the highest ratio at 77.4%, suggesting the strongest inclination for international market penetration. Denmark follows at 77.0% and France at 73.9%. This analysis provides insights into the extent of global rights acquisition.

Furthermore, analysis of the recent activity reveals that Denmark exhibits the highest research and development activity at 70.5%, followed by Japan at 61.7% and Korea at 55.6%. In terms of recent activity by applicant, Regeneron Pharma Inc. (Tarrytown, NY, USA) stands out with the highest activity at 85.6%, followed by Seagen Inc. (USA) and AbbVie Inc. (Chicago, IL, USA) at 68.8% and 66.3%, respectively (see Figure 9). This allows for an understanding of the research and development concentration based on the recent activity of key applicants.

### 3.4. Analysis of Key Technological Patent Portfolios in ADC Therapy

An analysis of technological capabilities based on the major patent portfolios of the world’s key patent applicants in ADC therapy was conducted. Firstly, Regeneron Pharma Inc. (USA) has filed a total of 4576 patents over 20 years, holding 132 patents in the field of ADC therapy, indicating a research and development concentration of 2.88% in the ADC therapy field. In the recent 5 years, they have filed 119 patents in the ADC therapy field, suggesting ongoing research and development and patent rights activities. The research and development concentration in the last 5 years is calculated at 57.98%. The calculation formula and the patent holding status of Regeneron Pharma Inc. are presented in Table 8.

Regeneron Pharma Inc. holds various patents related to ADC therapy, encompassing Verrucarin A derivatives and ADCs, protein–drug conjugates, Rifamycin analogs and ADCs, cyclodextrin protein–drug conjugates, and anti-MUC16 ADCs. These patents cover diverse drug payloads and conjugation technologies with specific antibodies. Regeneron Pharma Inc. has applied for or holds patents in the United States, Japan, Korea, and other countries for ADC formulations utilizing these technologies. The primary patents held by Regeneron Pharma Inc. are detailed in Table 9.

Looking at the patent portfolio of Seagen Inc. related to ADC therapy, they have filed a total of 167 patents over the past 20 years. In the field of ADC therapy, they hold 91 patents, indicating a technological concentration of 54.49%. Particularly noteworthy is their recent activity, with 76 patent applications in the ADC therapy field over the last 5 years, suggesting ongoing research and development efforts and active patenting. The degree of R&D concentration in the ADC therapy field for the recent 5-year period is calculated at 83.52% (see Table 10).

Examining the recent patents held by Seagen Inc. reveals a focus on various improvements and inventions related to different elements of ADC therapy. These include combination therapy, modulating the immune response, hydrophilic antibody–drug conjugates, B7-H4 ADCs, anti-PD-1 antibody in combination with anti-CD30 antibody, and β-glucuronidase-linker–drug conjugates. Seagen Inc.’s major patent list is provided in Table 11.

An analysis of the status of patent rights transfers has revealed a unique aspect of Seagen Inc. The company has established collaborative relationships with several research institutions for the development of ADC therapies. Seagen Inc. maintains cooperative research relationships through patent rights transfers with Seattle Genetics (USA) and Agensys Inc. (USA). The current status of patents related to collaborative research and joint research institutions for Seagen Inc. is presented in Table 12.

Next, we examined the patent portfolio of AbbVie Pharmaceuticals Inc. (USA), one of the major applicants, related to ADC therapies. AbbVie Pharmaceuticals has filed a total of 2851 patents over 20 years, possessing 170 patents in the field of ADC therapies, indicating a technological concentration of 5.96%. In particular, in the last 5 years, the number of patent applications in the ADC therapy field was 121, representing a recent research and development concentration of 71.18%. This suggests a notable focus on research and development in the field of ADC therapies (see Table 13).

We have also examined the recent technological content of AbbVie Pharmaceuticals’ patent holdings. The company has shown a focus on research and development, particularly in improving antibody technologies and drug conjugates. Specific areas of concentration include anti-EGFR antibody–drug conjugates, anti-c-Met ADCs, anti-EGFR ADC formulations, anti-CD98 ADCs, and anti-c-met ADCs. The key patent list held by AbbVie Pharmaceuticals is presented in Table 14.

Pfizer Inc. (USA), a company with multiple approved ADC therapy products and a stronghold on core strategic technologies in this field, has a significant patent portfolio. Over the last 20 years, Pfizer has filed 7555 patents, with 172 patents in the ADC therapy domain, indicating a technology concentration of 2.28%. Notably, in the recent five years, Pfizer has filed 60 patents in the ADC therapy domain, reflecting a recent R&D concentration of 34.88%. While the recent technology concentration may seem relatively low, Pfizer holds a substantial quantitative competitiveness and technological advantage. This suggests that Pfizer maintains a focus on research and development in the ADC therapy field (Table 15).

Analyzing the recent technological content of Pfizer’s patents reveals a concentration on the research and development of therapeutic antibodies and their applications, including cytotoxic peptides and antibody–drug conjugates, anti-PTK7 ADCs, bifunctional cytotoxic agents, therapeutic antibodies, spliceostatin analogs, and methods. Pfizer’s major patent holdings are presented in Table 16.

## 4. Conclusions and Future Perspectives

Antibody–drug conjugate (ADC) therapy emerges as a transformative technology, surmounting the constraints of traditional chemotherapy by leveraging targeted antibodies. Through extensive research and development, the fusion of diverse antibodies and cytotoxic payloads provides avenues for treating various cancer types. Presently, numerous pharmaceutical entities are leveraging ADC therapy to advance novel drugs, progressing through clinical trials. Over the last two decades, ADC therapy has witnessed remarkable strides, securing approvals for diverse cancer targets. The development of numerous ADC therapies, addressing hematologic and solid tumors, is reflected in the multitude of candidates currently undergoing clinical trials, pivotal for assessing efficacy and safety. The past five years have marked a substantial upswing in research and development concentration and patent filings concerning ADC therapy. Leading this technological frontier are notable patent applicants, including Pfizer Inc. (USA), AbbVie Pharmaceuticals Inc. (USA), Regeneron Pharma Inc. (USA), and Seagen Inc. (USA), steering advancements in the field. 

ADC technology, while brimming with promise for cancer treatment, faces significant challenges that impact its clinical success. These challenges encompass antigen expression discrepancies, tumor heterogeneity, off-target toxicities, immunogenicity, and pharmacokinetic hurdles. Effectively addressing these issues necessitates a multidisciplinary approach involving molecular biology, pharmacology, and clinical medicine. Ongoing research and innovations in ADC technology strive to overcome these obstacles by reinforcing antibody stability, optimizing payload selection, refining linker conjugation methods, and exploring synergies with nanotechnologies. Collaborative efforts across scientific disciplines are indispensable, playing a pivotal role in unlocking the full potential of ADCs in cancer treatment. This sustained collaboration aims to deepen our understanding of challenges and opportunities, ultimately driving advancements that elevate the efficacy and safety of ADCs in the battle against cancer.

## Figures and Tables

**Figure 1 pharmaceutics-16-00221-f001:**
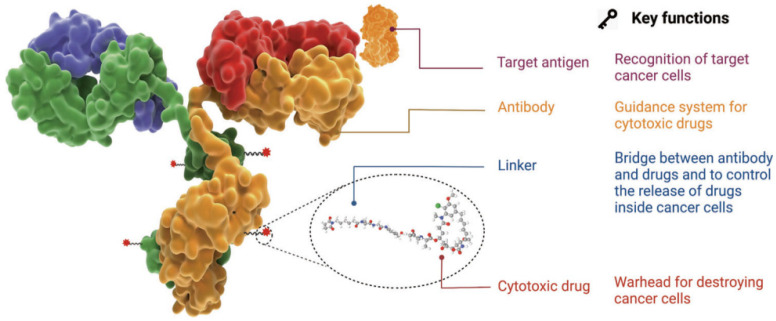
Characteristics and structure of ADC [1]. Copyright 2022 Springer Nature.

**Figure 2 pharmaceutics-16-00221-f002:**
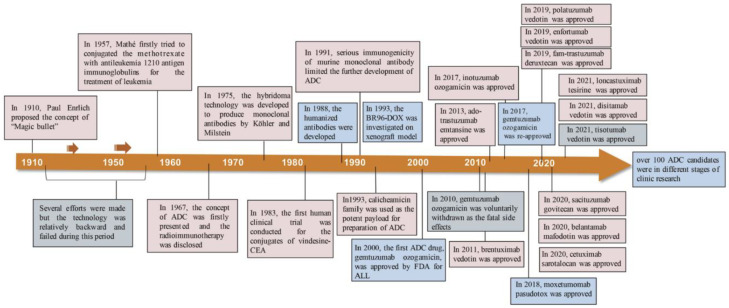
Timeline depicting important events in the development and approval of ADC drugs over the past century since the ‘magic bullet’ was proposed by Paul Enrlich in 1910 [1]. Copyright 2022 Springer Nature.

**Figure 3 pharmaceutics-16-00221-f003:**
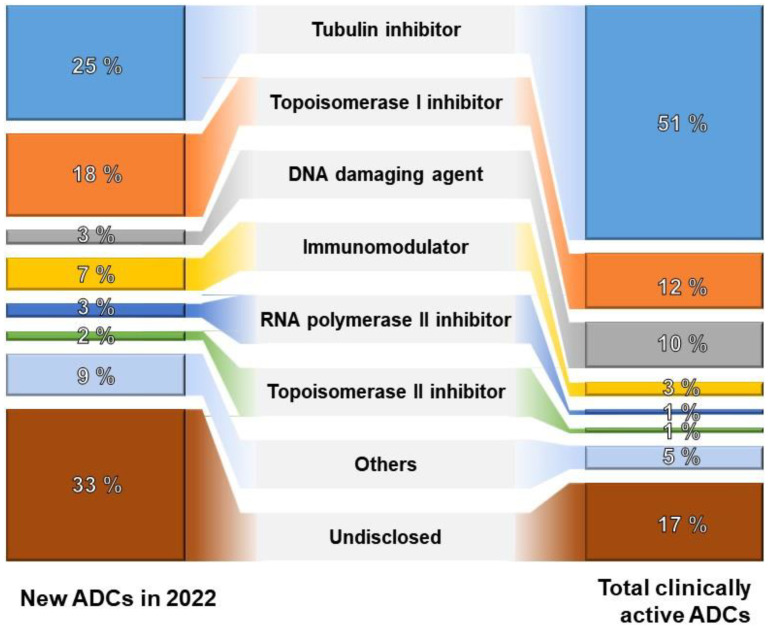
The proportion of ADCs by mechanism of action [67].

**Figure 4 pharmaceutics-16-00221-f004:**
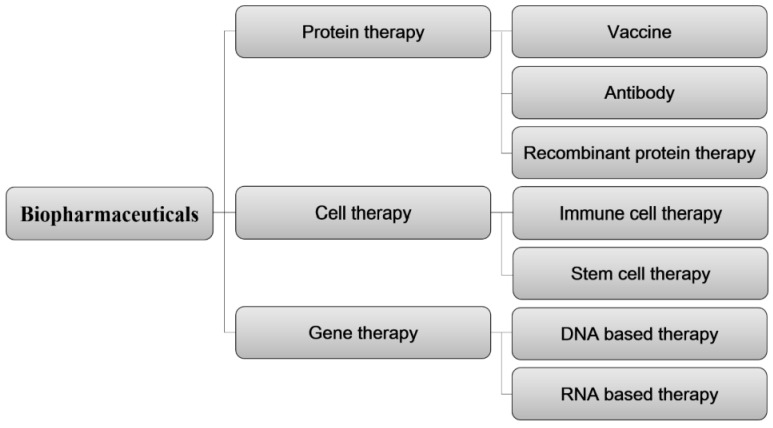
Classification of biopharmaceutical patent technology based on core technologies of biological materials.

**Figure 5 pharmaceutics-16-00221-f005:**
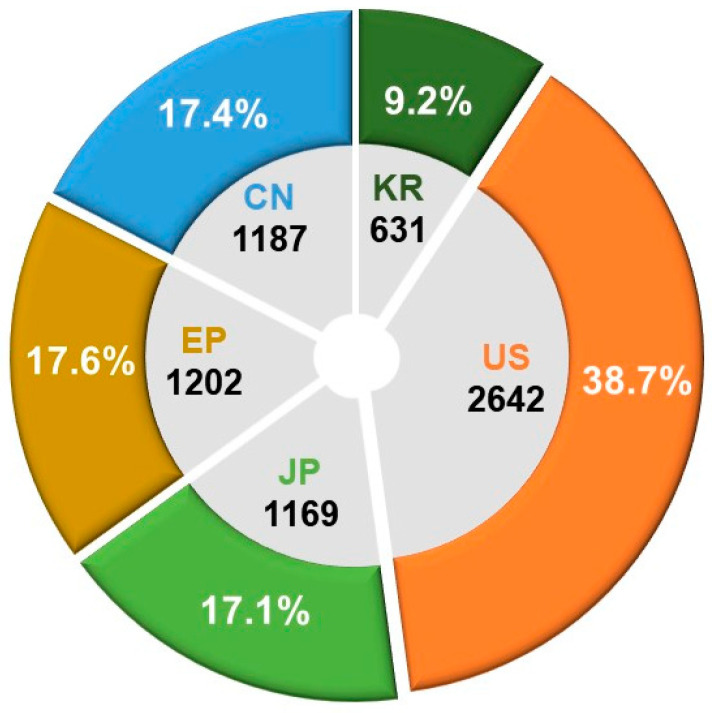
Patent office-specific counts and proportions of patent applications for biopharmaceutical technologies.

**Figure 6 pharmaceutics-16-00221-f006:**
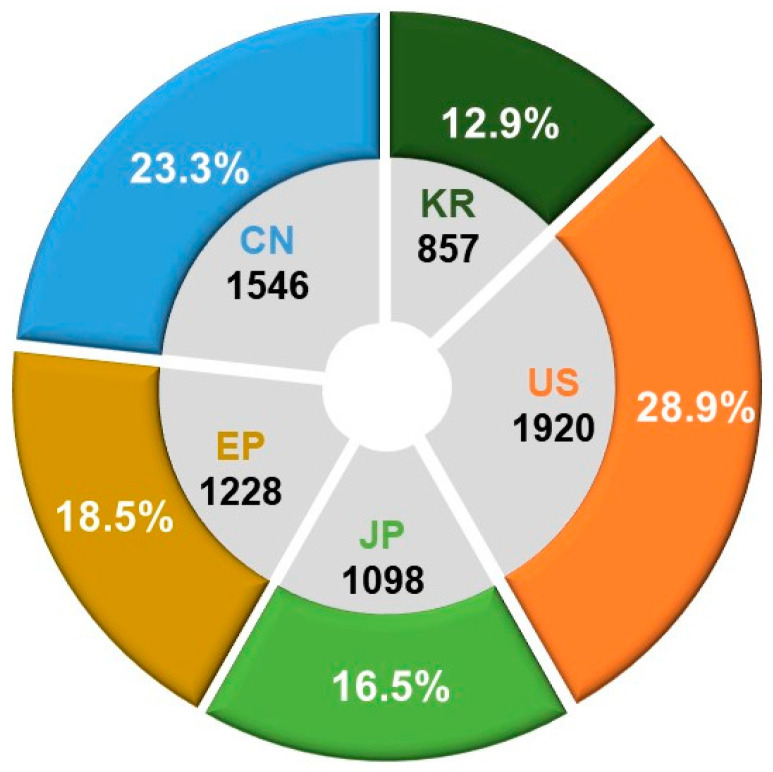
Patent office-specific counts and proportions of patent applications for ADC technologies.

**Figure 7 pharmaceutics-16-00221-f007:**
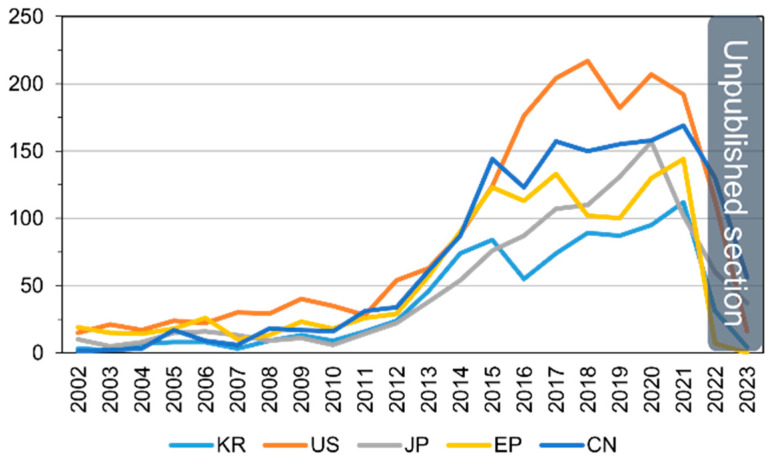
Patent application trends for ADC therapies by patent office.

**Figure 8 pharmaceutics-16-00221-f008:**
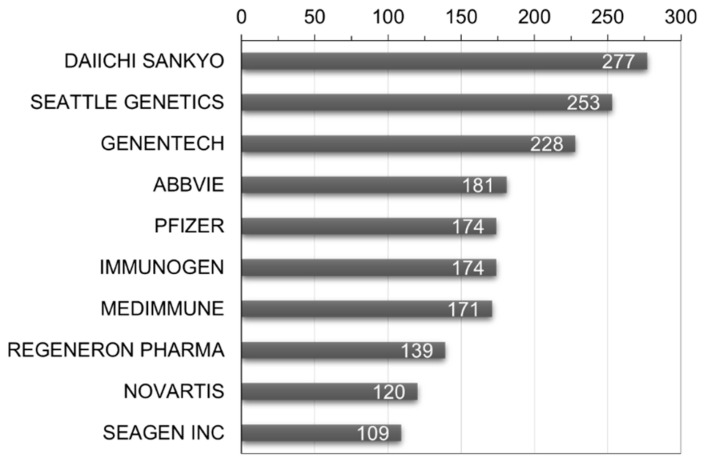
World’s major patent applicants and number of patent filings in ADC therapy.

**Figure 9 pharmaceutics-16-00221-f009:**
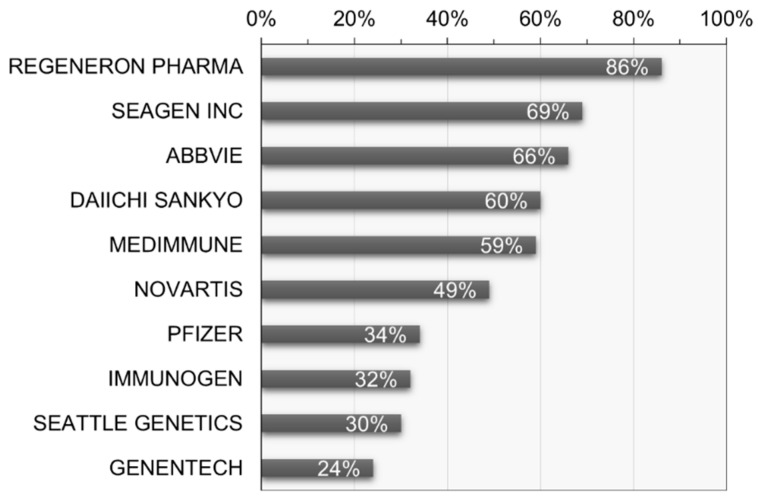
Recent research and development activity of key patent applicants in ADC therapy.

**Table 1 pharmaceutics-16-00221-t001:** Representative small molecular cytotoxic payloads [19].

Category	Name	Structure	Mechanism	ADC Drug
**Tubulin inhibitors**	Auristatins	MMAE, MMAF	Promote tubulin Polymerization and target at the β-subunits of tubulin dimer to perturb microtubule growth	Brentuximab Vedotin Polatuzumab Vedotin Enfortumab Vedotin Belantamab Mafodotin Disitamab Vedotin Tisotumab Vedotin
	Maytansinoids	Mertansine, DM1 Ravtansine, DM4	Block the polymerization of tubulin dimer and inhibit the formation of mature microtubules	Trastuzumab Emtansine
	Tubulins	Tubulysin A	Inhibit tubulin polymerization	
**DNA-damaging agents**	Calicheamicins	Calicheamicin-gamma1	DNA double strand break	Gemtuzumab Ozogamicin Inotuzumab Ozogamicin
	Duocarmycins	Duocarmycin	DNA alkylation	
	Exatecans	DXd	Topoisomerase I inhibitor	Trastuzumab Deruxtecan Sacituzumab Govitecan
	Pyrrolobenzodiazepines	PBD	Crosslinking of DNA	Loncastuximab Tesirine

ADC, antibody–drug conjugate; MMAE, monomethyl auristatin E; MMAF, monomethyl auristatin F; DM1, emtansine; DXd, deruxtecan; SN-38, 7-ethyl-10-hydroxy-camptothecin; PBD, pyrrolobenzodiazepine.

**Table 2 pharmaceutics-16-00221-t002:** ADC drugs approved for hematologic malignancies [49].

ADC	Manufacturers	Approval Years	Targets	Payloads	Indications
Adcetris (brentuximab vedotin)	Seagen	2011	CD30	Monomethyl auristatin E (MMAE)	Adults with classical Hodgkin’s lymphoma (CHL) in stages 3 and 4 without prior treatment history Pediatric patients aged 2 and above with high-risk classical Hodgkin’s lymphoma (CHL) and no previous treatment history Adult patients with systemic anaplastic large cell lymphoma (ALCL) without a history of prior treatment Adult patients with primary cutaneous anaplastic large cell lymphoma
Blenrep (belantamab mafodotin)	GSK	2020	BCMA	Monomethyl auristatin F	Adult patients with relapsed or refractory multiple myeloma who have received the selected treatment at least four times
Besponsa (inotuzumab ozogamicin)	Pfizer	2017	CD22	Calicheacmicin	Monotherapy for relapsed or refractory B-cell precursor acute lymphoblastic leukemia (ALL) in adults
Mylotarg (gemtuzumab ozogamicin)	Pfizer	2000; 2017	CD33	Calicheacmicin	Monotherapy and combination therapy (up to 2 years) for newly diagnosed CD33-positive AML in adults and relapsed or refractory CD33-positive AML in both adults and pediatric patients
Polivy (polatuzumab vedotin)	Genentech	2019	CD79b	MMAE	Combination therapy with bendamustine and rituximab in adult patients with relapsed or refractory indolent B-cell lymphoma
Zynlonta (loncastuximab tesirine)	ADC Therapeutics	2021	CD19	SG3249 PBD dimer	Adult patients with relapsed or refractory diffuse large B-cell lymphoma (DLBCL), including DLBCL arising from low-grade lymphoma and high-grade B-cell lymphoma, after two or more systemic therapies, not otherwise specified

**Table 3 pharmaceutics-16-00221-t003:** ADC drugs approved for solid tumors [49].

ADC	Manufacturers	Approval Years	Targets	Payloads	Indications
Kadcyla (adorastuzumab emtansine)	Roche/Genentech	2013	HER2	DM1	A single agent for the treatment of HER2-positive metastatic breast cancer (MBC) patients who have previously received trastuzumab and taxane (separately or in combination). Additionally, a single-agent adjuvant therapy for HER2-positive early breast cancer patients.
Enhertu (trastuzumab deruxtecan)	Daichi Sankyo/AstraZeneca	2019	HER2	Exatecan-derivative topoisomerase I inhibitor (DXd)	Adult patients with inoperable or metastatic HER2-positive breast cancer. Adult patients with inoperable or metastatic HER2-low breast cancer. Adult patients with inoperable or metastatic non-small cell lung cancer (NSCLC) harboring an activating HER2 (ERBB2) mutation. Treatment for adult patients with locally advanced or metastatic HER2-positive gastric or gastroesophageal junction (GEJ) adenocarcinoma.
Trodelvy (sacituzumab govitecan)	Immunomedics/Gi lead	2020	Trop-2	SN-38 Topo 1 inhibitor	Adult patients with metastatic triple-negative breast cancer (mTNBC) who have received at least two prior treatments for metastatic disease.
Padcev (enfortumab vedotin)	Seagen/Astellas Pharma	2019	Nectin-4	MMAE	Adult patients with locally advanced or metastatic urothelial carcinoma (UC) who have received prior programmed cell death receptor-1 (PD-1) or programmed death-ligand 1 (PD-L1) inhibitor therapy, as well as platinum-containing chemotherapy in the neoadjuvant or adjuvant setting, and have received at least two prior systemic regimens for advanced or metastatic disease.
Tivdak (tisotumab vedotin)	Genmab/Seagen	2021	TF-011	MMAE	Recurrent or metastatic cervical cancer progressing during or after chemotherapy.
Elahere (mirvetuximab soravtansine)	ImmunoGen	2022	FOLR1	Maytansinoid DM4	Adult patients with folate receptor alpha-positive, platinum-resistant epithelial ovarian, fallopian tube, or primary peritoneal cancer, who have previously received 1~3 systemic treatment regimens.

**Table 4 pharmaceutics-16-00221-t004:** The majority of ADC developers and their collaborations [49,56,57,58,59,60].

Date	Company 1	Company 2	Type of Collaboration	Estimated Value	Details
January 2022	ADC Therapeutics S.A.	Mitsubishi Tanabe Pharma Corporation	License agreement	USD 230 million	ADC Therapeutics has entered into an exclusive licensing agreement for the development and commercialization of ZYNLONTA (loncastuximab tesirine).
February 2022	Mersana Therapeutics	Janssen Biotech	Collaboration	USD 1 billion	The collaboration aims to leverage Janssen’s exclusive antibodies and Mersana’s expertise in ADCs, combining them with the Dolasynthen platform and targeting three specific objectives to discover new ADCs. Mersana may utilize Synaffix’s GlycoConnect™ technology with its site-selective ADC bioconjugation approach.
February 2022	Abzena	OBI Pharma: Odeon Therapeutics	License agreement	USD 200 million	OBI has granted rights to ADCOBI-999 targeting Globo H (a glycosphingolipid antigen expressed in 15 types of cancer, including breast, prostate, lung, stomach, esophageal, and colorectal cancers).
May 2022	Byondis	Medac GmbH	Collaboration	Confidential	Byondis has entered into a licensing, collaboration, and supply agreement for the anti-HER2 ADC Trastuzumab Duocarmazine (SYD985) with Medac. Medac obtains an exclusive license to commercialize SYD985 for all approved indications in the European Union, the United Kingdom, and other European countries.
May 2022	LaNova Medicines Limited	Turning Point Therapeutics	License agreement	USD 120 million	Turning Point Therapeutics has announced a licensing agreement for the clinical-stage anti-Claudin18.2 ADC TPX-4589 (LM-302) for gastrointestinal cancer with Lanova Medicines as part of its pipeline expansion.
June 2022	Boehringer Ingelheim	Agency for Turning Point Therapeutics Technology and Research	License agreement	USD 1.14 billion	Boehringer Ingelheim plans to globally manufacture and sell customized cancer therapies using A*Star’s proprietary antibodies. Through this collaboration, Boehringer Ingelheim can enhance its portfolio of immune cell-targeted (T-cell engagers) and tumor cell-targeted ADCs (antibody–drug conjugates).
June 2022	Exelixis	BioInvent	License agreement	Confidential	Exelixis and BioInvent have entered into an exclusive option and license agreement to create cutting-edge immune-oncology therapies based on a new antibody. Exelixis will make efforts towards all future research and commercialization for development, including promising ADC and bispecific antibody engineering technologies, and plans to pay BioInvent an option exercise fee.
October 2022	Synaffix B.V.	Emergence Therapeutics A.G.	License agreement	USD 360 million	Synaffix and Emergence have entered into a licensing agreement providing Emergence access to target-specific-based Synaffix exclusive ADC technology comprising GlycoConnect™, HydraSpace™, and SYNtecan E™ linkers as payloads.
December 2022	Amgen	LegoChem Biosciences. Inc. (LCB)	License agreement	USD 1.25 billion	This deal includes a multi-target research collaboration and a licensing agreement for ADC development.
December 2022	Merck KGaA	Mersana Therapeutics	License agreement	A prepaid payment of USD 30 million, with an additional USD 800 million in global sales royalties and milestone payments	The developed ADC activates the STING (Stimulator of Interferon Gene) signaling pathway, stimulating the immune system and causing the destruction of cancer cells through this immune antitumor action.
December 2022	Merck & Co.	Kelun-Biotech	License agreement	An upfront payment of USD 175 million and a global milestone payment of up to USD 930 million for generating seven ADC candidates for royalties	The Chinese company Kelun-Biotech has granted a global license to Merck & Co. to generate seven ADC candidates for cancer treatment. Kelun retains the rights to research, generate, and sell specific ADCs for its country with additional payments in dollars.
April 2023	BioNTech SE	DualityBio	License agreement	An upfront payment of USD 170 million, with additional royalties and milestones totaling over USD 1.5 billion	For two ADC assets from DualityBio, BioNTech has entered into a global (excluding certain Asian regions) exclusive license and partnership agreement for development and commercialization.
July 2023	BeiGene	DualityBio	License agreement	USD 13 billion	Licensing agreement to advance differentiated antibody–drug conjugate (ADC) therapy for solid tumors.
October 2023	GSK	Hansoh	License agreement	An upfront payment of USD 185 million, with success-based milestones totaling up to USD 1.525 billion	Licensing agreement for HS-20093, a compound with top-tier potential in ovarian and endometrial cancers and promising possibilities in other solid tumors.
October 2023	Endeavour BioMedicines	Hummingbird Bio	License agreement	USD 430 million	Licensing agreement for HMBD-501, the next-generation HER3-targeted antibody–drug conjugate (ADC) with an optimized taxane payload for enhanced safety and efficacy.
December 2023	Pfizer	Nona Biosciences	License agreement	An upfront payment of USD 53 million, with success-based milestones totaling up to USD 1.05 billion	Licensing agreement for HBM9033, a mesothelin-targeted antibody–drug conjugate (ADC) for solid tumors.
December 2023	BMS	SystImmune	License agreement	An upfront payment of USD 800 million, with a potential maximum of USD 8.4 billion	Licensing agreement for BL-B01D1, a promising candidate for the treatment of EGFR-mutant non-small cell lung cancer (NSCLC).

**Table 5 pharmaceutics-16-00221-t005:** Patent landscapes of 11 FDA-approved ADC drugs [61].

ADC	Linker	Cytotoxin	Linker and Cytotoxin	Formula
Mylotarg (gemtuzumab ozogamicin)	Linkers containing acylhydrazide and thiol moieties conjugated to the calicheamicin analog (US5606040 filed in 1993)	Calicheamicin analog (US5053394 filed in 1989, US5079233 filed in 1989)	Ozogamicin (US5773001 filed in 1994)	US5712374 filed in 1995, US5714586 field in 1996
Adcetris (brentuximab vedotin)	MC-VC-PABC linker (US6214345 filed in 1993)	MMAE (US6884869 filed in 2001)	Vedotin (US7659241 filed in 2003)	US7659241 filed in 2003
Kadcyla (trastuzumab emtansine)	p-carboxycyclo hexylmethylmaleimide linker (JP52085164 filed in 1977)	Maytansine (US3896111 filed in 1973) Mertansine/DM1 (US5208020 filed in 1992, US5416064 filed in 1992)	Emtansine (US5208020 filed in 1992)	US7097840 filed in 2001, US8088387 filed in 2004
Besponsa (inotuzumab ozogamicin)	Linkers containing acylhydrazide and thiol moieties conjugated to the calicheamicin analog (US5606040 filed in 1993)	Calicheamicin analog (US5053394 filed in 1989, US5079233 filed in 1989)	Ozogamicin (US5773001 filed in 1994)	US8153768 filed in 2003, US8835611 filed in 2012, US9351986 filed in 2014
Polivy (polatuzumab vedotin)	MC-VC-PABC linker (US6214345 filed in 1993)	MMAE (US6884869 filed in 2001)	Vedotin (US7659241 filed in 2003)	US8088378 filed in 2008
Padcev (enfortumab vedotin)	MC-VC-PABC linker (US6214345 filed in 1993)	MMAE (US6884869 filed in 2001)	Vedotin (US7659241 filed in 2003)	US8637642 filed in 2011
Enhertu (trastuzumab deruxtecan)	GGFG linker (US6436912 filed in 1997, US5688931 filed in 1994)	Camptothecin derivative (US5658920 filed in 1995)	Deruxecan (US10195288 filed in 2013)	US10155821 filed in 2016
Trodelvy (sacituzumab govitecan)	CL2A linker (US8420086 filed in 2011)	7-ethyl-10-hydroxycamptothecin (SN-38) (US4473692 filed in 1982) SN-38 with PABC spacer (US8877901 filed in 2006)	Govitecan (US8420086 filed in 2011)	US9028833 filed in 2013
Blenrep (belantamab mafodotin)	MC linker	MMAF (US7662387 filed in 2004)	Mafodotin (US7498298 filed in 2004)	US9273141 filed in 2013
Zynlonta (loncastuximab tesirine)	MP-PEG8-VA-PABC linker (US20110256157 filed in 2011)	Dimeric PBDs (US7049311, US7067511, US7265105 filed in 1999)	Tesirine (US9889207 filed in 2013)	US9931414 filed in 2013
Tivdak (tisotumab vedotin)	MC-VC-PABC linker (US6214345 filed in 1993)	MMAE (US6884869 filed in 2001)	Vedotin (US7659241 filed in 2003)	US10617764 filed in 2014

MC, maleimide caproyl; VC, valine citrulline; PABC, p-aminobenzyloxycarbamoyl; GGFG, glycine-glycine-phenylalanine-glycine; MP, maleimide propoyl; PEG8, eight polyethylene glycol; VA valine-alanine; MMAE, monomethyl auristatin E; MMAF, monomethyl auristatin F; PBD, pyrrolobenzodiazepine.

**Table 6 pharmaceutics-16-00221-t006:** Core technologies of biopharmaceutical patents and search keywords.

Core Technologies	Search Keywords and CPC Classifications
Biopharmaceuticals: vaccine, antibody, recombination protein, cell	**Search keywords:** (((Chimeric Mutant* Mutation* Mutagenesis* Transgenic Variant* Exogenous Engineer* Recombin* Transfection*) AND (Protein* (Poly ADJ2 peptide*) Ferritin* Peptid Nucleotid*)) OR (Vaccin* Pneumoco* Antibody* (Immuno ADJ2 globulin*) Antibodies* (Anti ADJ2 bod*) Autoantibody* Antagonist*)) OR ((Adipo* Fat*) AND (Stem*) AND (Cell*)) **CPC technology classifications:** A61K48/00, A61P35/00, A61K48/005

*, truncation symbol for instructing the database to search for all forms of a word.

**Table 7 pharmaceutics-16-00221-t007:** Core technologies of ADC patents and search keywords.

Core Technologies	Search Keywords and CPC Classifications
ADC: antibody–drug conjugate, linker, payloads	Search keywords: (antibody drug conjugate), ((payload OR drug OR cytotoxic) AND (linker OR bridge OR site specific)) CPC technology classifications: A61P35/00, A61K47/6803, A61K47/6849

**Table 8 pharmaceutics-16-00221-t008:** Regeneron Pharma Inc. (USA) patent portfolio and degree of technology and R&D concentration.

Target Technology (ADC Therapy)				Corporate Patent Holdings		
The number of patents in the last 20 years (A)	The number of patents in the last 5 years (B)	Degree of technology concentration in the last 20 years (%) (C = A/E)	Degree of recent concentration in the last 5 years (%) (D = B/A)	Total application number in the last 20 years (E)	Application number in the last 5 years (F)	Degree of R&D concentration in the last 5 years (%) (G = F/E)
132	119	2.88	90.15	4576	2653	57.98

**Table 9 pharmaceutics-16-00221-t009:** List of ADC-therapy-related patents held by Regeneron Pharma Inc. (USA).

No.	Territory	Application No.	Filing Date	Title	Status
1	US	US18/154262	2023-01-13	Verrucarin a derivatives and antibody drug conjugates thereof	Filing
2	US	US18/095467	2023-01-10	Protein-drug conjugates comprising camptothecin analogs and methods of use thereof	Filing
3	US	US18/090138	2022-12-28	Rifamycin analogs and antibody-drug conjugates thereof	Filing
4	US	US17/953114	2022-09-26	Anti-MUC16 antibodies, antibody-drug conjugates, and bispecific antigen-binding molecules that bind MUC16 and CD3, and uses thereof	Filing
5	US	US17/837598	2022-06-10	Cyclodextrin protein drug conjugates	Filing
6	JP	JP2022063698A	2022-04-07	Anti-steap2 antibody, antibody-drug conjugate, and bispecific antigen binding molecule that binds steap2 and CD3, and use of the same	Filing
7	US	US17/666116	2022-02-07	Antigen binding molecule formats	Registered
8	US	US17/454186	2021-11-09	Selenium antibody conjugates	Filing
9	KR	KR20237017674A	2021-11-09	Selenium antibody conjugates	Filing
10	CN	CN202180078123A	2021-11-09	Selenium antibody conjugates	Filing

**Table 10 pharmaceutics-16-00221-t010:** Seagen Inc. (USA) patent portfolio and degree of technology and R&D concentration.

Target Technology (ADC Therapy)				Corporate Patent Holdings		
The number of patents in the last 20 years **(A)**	The number of patents in the last 5 years **(B)**	Degree of technology concentration in the last 20 years (%) **(C = A/E)**	Degree of recent concentration in the last 5 years (%) **(D = B/A)**	Total application number in the last 20 years **(E)**	Application number in the last 5 years **(F)**	Degree of R&D concentration in the last 5 years (%) **(G = F/E)**
91	76	54.49	83.52	167	139	83.23

**Table 11 pharmaceutics-16-00221-t011:** List of ADC-therapy-related patents held by Seagen Inc. (USA).

No.	Territory	Application No.	Filing Date	Title	Status
1	US	US18/309388	2023-04-28	Combination therapy	Filing
2	US	US18/047240	2022-10-17	Modulating the immune response using antibody-drug conjugates	Filing
3	US	US18/047243	2022-10-17	Combination therapy using a liv1-adc and a chemotherapeutic	Filing
4	US	US17/963939	2022-10-11	Hydrophilic antibody-drug conjugates	Filing
5	US	US17/936814	2022-09-29	B7-h4 antibody-drug conjugates for the treatment of cancer	Filing
6	JP	JP2022130733A	2022-08-18	Use of anti-pd-1 antibody in combination with anti-cd30 antibody in lymphoma treatment	Filing
7	US	US17/819819	2022-08-15	Anti-NTB-A antibodies and related compositions and methods	Filing
8	JP	JP2022117254A	2022-07-22	β-glucuronide-linker drug conjugate	Filing
9	US	US17/870749	2022-07-21	Use of antibody drug conjugates comprising tubulin disrupting agents to treat solid tumor	Filing
10	US	US17/835841	2022-06-08	Process for the preparation of tubulysins and intermediates thereof	Filing

**Table 12 pharmaceutics-16-00221-t012:** Seagen Inc.’s joint research patents and collaborative research institutions related to ADC therapies.

No.	Application No.	Title	Collaborative Institutions
1	US16/934745	Humanized anti-liv1 antibodies for the treatment of cancer	Seattle Genetics
2	EP21870200A	Methods for treating cancers with antibody drug conjugates (ADC) that bind to 191p4d12 proteins	Agensys
3	US17/779068	Treatment of cancers with antibody drug conjugates (ADC) that bind to 191p4d12 proteins	Agensys
4	EP21878582A	Methods for treating cancers with antibody drug conjugates (ADC) that bind to 191p4d12 proteins	Agensys
5	EP21827092A	Markers for use in methods for treating cancers with antibody drug conjugates (ADC)	Agensys

**Table 13 pharmaceutics-16-00221-t013:** AbbVie Pharmaceuticals Inc. (USA) patent portfolio and degree of technology and R&D concentration.

Target Technology (ADC Therapy)				Corporate Patent Holdings		
The number of patents in the last 20 years **(A)**	The number of patents in the last 5 years **(B)**	Degree of technology concentration in the last 20 years (%) **(C = A/E)**	Degree of recent concentration in the last 5 years (%) **(D = B/A)**	Total application number in the last 20 years **(E)**	Application number in the last 5 years **(F)**	Degree of R&D concentration in the last 5 years (%) **(G = F/E)**
170	121	5.96	71.18	2851	758	26.59

**Table 14 pharmaceutics-16-00221-t014:** List of ADC-therapy-related patents held by AbbVie Pharmaceuticals Inc. (USA).

No.	Territory	Application No.	Filing Date	Title	Status
1	JP	JP2023025869A	2023-02-22	Anti-EGFR antibodies and antibody drug conjugates	Filing
2	JP	JP2022152185A	2023-09-26	Anti-c-met antibody drug conjugates and methods for their use	Filing
3	JP	JP2022098605A	2022-06-20	Anti-EGFR antibody drug conjugate formulations	Filing
4	US	US17/806037	2022-06-08	Anti-CD98 antibodies and antibody drug conjugates	Filing
5	EP	EP22796992A	2022-04-29	Anti-c-Met antibody drug conjugates	Filing
6	US	US17/661450	2022-04-29	Anti-c-Met antibody drug conjugates	Registered
7	JP	JP2022530214A	2022-04-29	Anti-c-Met antibody drug conjugates	Filing
8	US	US17/731226	2022-04-27	Anti-CD98 antibodies and antibody drug conjugates	Filing
9	US	US17/688908	2021-03-08	Anti-EGFR antibody drug conjugates	Filing
10	KR	KR20237026876A	2021-01-20	Anti-EGFR antibody drug conjugates	Filing

**Table 15 pharmaceutics-16-00221-t015:** Pfizer Inc. (USA) patent portfolio and degree of technology and R&D concentration.

Target Technology (ADC Therapy)				Corporate Patent Holdings		
The number of patents in the last 20 years **(A)**	The number of patents in the last 5 years **(B)**	Degree of technology concentration in the last 20 years (%) **(C = A/E)**	Degree of recent concentration in the last 5 years (%) **(D = B/A)**	Total application number in the last 20 years **(E)**	Application number in the last 5 years **(F)**	Degree of R&D concentration in the last 5 years (%) **(G = F/E)**
172	60	2.28	34.88	7555	1314	17.39

**Table 16 pharmaceutics-16-00221-t016:** List of ADC-therapy-related patents held by Pfizer Inc. (USA).

No.	Territory	Application No.	Filing Date	Title	Status
1	US	US14/793942	2015-07-08	Cytotoxic peptides, and antibody drug conjugates thereof	Registered
2	KR	KR20207013177A	2016-03-30	Therapeutic antibodies and their uses	Registered
3	KR	KR20207013178A	2016-03-30	Therapeutic antibodies and their uses	Registered
4	US	US14/696663	2015-04-27	Anti-PTK7 antibody-drug conjugates s	Registered
5	US	US14/605697	2015-01-26	Bifunctional cytotoxic agents	Registered
6	US	US14/818455	2015-08-05	Spliceostatin analogs and methods for their preparation	Registered
7	US	US16/017974	2018-06-25	Therapeutic antibodies and their uses	Registered
8	US	US16/053344	2018-08-02	Bifunctional cytotoxic agents	Registered
9	KR	KR20177032304A	2016-03-30	Therapeutic antibodies and their uses	Registered
10	US	US15/610417	2017-05-31	Anti-CXCR4 antibodies and antibody-drug conjugates	Registered

## Data Availability

Not applicable.

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
