# Peer review of "Recent Technological and Intellectual Property Trends in Antibody–Drug Conjugate Research"

_pharmaceutics, 2024, doi:10.3390/pharmaceutics16020221_

Round 1
Reviewer 1 Report
Comments and Suggestions for Authors
Youngbo Choi and colleagues present a thorough examination of the evolving landscape of ADC therapy. The paper meticulously explores the development of critical components like antibodies, linkers, and cytotoxic payloads. It sheds light on the increasing global investment in anti-cancer agents and the expansion of the ADC therapy market, while thoughtfully addressing the challenges and prospective advancements in this sector. The authors have done a commendable job in discussing the surge in clinical trials and research, with a particular emphasis on the pivotal contributions from leading companies. This work offers an in-depth view of the technological progress, market trends, and patent landscapes within the realm of ADC therapy. While the paper is well-structured and insightful, I recommend acceptance with minor revisions. The authors could enhance the paper by:
- Providing a more detailed comparison between currently approved drugs and those in clinical trials, with a focus on the advantages and limitations of present ADC therapies.
- Incorporating a case study of successful ADC therapies to exemplify the practical application of the discussed concepts.
- Including a dedicated section on state-of-the-art ADC technologies, such as probody drug conjugates, to underscore recent innovations in the field.
- Expanding the scope of the patent landscape analysis to offer more granular details about key patents and their influence on ADC therapy.
- In the concluding discussion, elaborating on future predictions and potential challenges in ADC therapy. This should include insights from related disciplines like biotechnology or pharmacoeconomics to provide a more rounded, interdisciplinary perspective.
Author Response
We sincerely appreciate your thorough review and valuable comments. Your insights have proven instrumental in enhancing the overall quality of our research. We have diligently incorporated revisions in accordance with the reviewer's feedback. Thank you sincerely for your time and constructive input.

Reviewer 2 Report
Comments and Suggestions for Authors
1. The title is not appropriate since most of this manuscript is about basic background (which has been summarized in many other publications), market analysis, and patent application analysis.
2. All of the cited references are review papers or reports. The manuscript failed to provide a "scientific review" but more like a summary of review papers/patent filing reports.
3. The whole manuscript is based on one assumption: if a company files more patent applications means the company is more technologically capable. The author ignored that a patent application can be withdrawn, or will not be granted eventually. Additionally, to gain global rights, one usually files the same patent in many countries, e.g. in Table 8, No. 8, 9, and 10, are these repetitive applications screened or counted as 3 activities? In short, the patent application numbers may not necessarily mean technological advances. The direct indication could be more and more people are working or have worked on it. Moreover, usually, there are 18 months before the publication of a patent application. So there the trend is 18 months ago.
4. The manuscript simply listed the patent titles without any insight into the ADC technology details described in the patents. Also, a little bit more insight into the application of the patents would be appreciated by the readers. For example, some of the listed ADC patents already failed in clinical trials and further development has been suspended. How do cases like those impact the ADC field?
5. Indeed, ADC is a hot area as of now. The author simply listed patent filing status analysis with hardly scientific interpretation and failed to provide technology-related viewpoints. The manuscript would not be suitable for publication in Pharmaceutics, and will not be helpful for the current ADC field.
Other minors:
1. Line 116, lysine conjugates are not necessarily unstable. The statement could be misleading, it could be chemistry stability, physical stability, serum stability, etc. Please revise more precisely.
2. Line 121, site-specific conjugation is not necessarily more stable and facilitates drug release, please be precise.
3. In Table 3, Enhertu's target should be HER2.
4. In Table 4, the table title is wrong. Also, include recent noteworthy partnerships, e.g., GSK, BMS, etc.
5. Line 204-206, the ILD is unrelated to the HER2 target.
6. Line 208-252, "Worldwide biopharmaceutical patent trends" This part does not contribute much to the topic.
7. Table 6, the title is inappropriate for the table content.
Comments on the Quality of English Language
Minor English editing is needed.
Author Response

(The authors gave the same response as above.)

Reviewer 3 Report
Comments and Suggestions for Authors
I have thoroughly reviewed your comprehensive review article on antibody-drug conjugates (ADCs), focusing on technical aspects and trends in intellectual property. Your work is well-written and provides valuable insights into the field. However, I would like to suggest several areas for enhancement to further enrich your manuscript.
Regarding Conjugation Technology: Your discussion on chemical conjugation is informative, yet it would be beneficial to expand this section to include enzymatic methods, which are emerging as a significant trend. Recent developments in tag-free antibody modification techniques are noteworthy and would enrich your review. Including these methods would provide a more complete overview of the current state of ADC conjugation technology.
Concerning Payloads: The title of your paper emphasizes 'cytotoxic' payloads, but recent advancements have seen the emergence of non-cytotoxic payloads like STING and TLR agonists. I recommend revising this section to reflect these developments. Additionally, addressing novel format conjugates such as oligonucleotides, PROTACs, and dyes would provide a more comprehensive understanding of the evolving landscape of ADC payloads.
On Patent Strategies: An exploration of patent strategies in the realm of ADCs would significantly add to the depth of your review. ADCs encompass a wide range of patents, from antibodies and payloads to the ADCs themselves. Discussing the strategies for extending patent life, such as filing process patents for life cycle management, would offer valuable insights into the intellectual property aspects of ADC development. Including examples of such life cycle management strategies would be particularly beneficial.
Regarding References: Some of the review articles cited (e.g., Ref 7, 9, 11, 23) appear outdated. An update to more recent and relevant references, especially for those like Ref 23, would strengthen your review. Ref 23, focusing on early-stage conjugation methods, might be less relevant to the current practices in ADC development. Replacing it with literature that details chemical conjugation methods used in more recent, practical ADC applications would be advisable.
Author Response

(The authors gave the same response as above.)

Round 2
Reviewer 2 Report
Comments and Suggestions for Authors
The revision has addressed most of the questions.
Line 296-298, what are the results of the early clinical trials, for example, XMT-2056, SBT6050? And can the clinical results be interpreted as "promising" or "great potential"? Please cite related references.
Other minors:
Table 5 and related context, all the term "formulation" should be "formula". Enhertu's cytotoxin is not Exatecan. Below the table "GCFG" should be GGFG, and "Monomethyl" should be "monomethyl".
Line 288, missing "agonist" after (STING).
Comments on the Quality of English LanguageMinor editing of English language required
Author Response
Thank you for your thoughtful review and valuable feedback. Your comments have proven to be immensely beneficial in enhancing the overall quality of our research. We have carefully incorporated the suggested revisions based on the reviewer's comments. We sincerely appreciate your time and commitment to ensuring the excellence of our work. Once again, thank you for your invaluable input.

Reviewer 3 Report
Comments and Suggestions for Authors
The authors have successfully addressed the majority of the comments raised in the previous review round. However, I recommend further enhancement in the clarity and coherence of the manuscript.
Specifically, I suggest the removal of Reference 23 by Yamada, K. et al., as its content appears to be outdated and its key points are largely encompassed by References 24, 29-31. To supplement this, I propose incorporating a more recent review that discusses tag-free enzymatic conjugation, which can be found at https://doi.org/10.1002/slct.202203753.
This addition could provide a more contemporary perspective and enrich the review article's relevance to current research in the field.
Author Response

(The authors gave the same response as above.)
